

# Uncertainties in OCO-2 satellite retrievals of XCO₂ limit diagnosis of transport model simulation uncertainty

Chiranjit Das[1], Ravi Kumar Kunchala[1], Prabir K. Patra[2,3], Naveen Chandra[2], Kentaro Ishijima[4], Toshinobu Machida[5]

[1] Centre for Atmospheric Sciences, Indian Institute of Technology Delhi, New Delhi, India

[2] Japan Agency for Marine-Earth Science and Technology (JAMSTEC), Yokohama 236-0001, Japan

[3] Research Institute for Humanity and Nature, Kyoto 6038047, Japan

[4] Meteorological Research Institute, Tsukuba, Japan

[5] National Institute for Environmental Studies, Tsukuba, Japan

**Correspondence to**: Prabir K. Patra (prabir@jamstec.go.jp) and Chiranjit Das (Chiranjit.Das@cas.iitd.ac.in)

**Abstract.** Estimating regional $CO_2$ sources and sinks is challenging due to limited data and uncertainties in transport models. Orbiting Carbon Observatory-2 (OCO-2) overcomes measurement limits, providing $CO_2$ variations beyond in-situ networks. This study analyses altitude-wise model-observation $CO_2$ differences from surface to upper troposphere using aircraft observations from ATom, Amazon, and CONTRAIL campaigns over OCO-2 total column $CO_2$ ($XCO_2$) sampling location to characterise sources of uncertainty in MIROC4-ACTM. We show model aligns better with ATom tropospheric columns ($0.03 \pm 0.03$ ppm) than OCO-2 $XCO_2$ ($0.2 \pm 0.5$ ppm), especially over oceans, highlighting the need for expanded profile measurements to characterise errors robustly. Altitude-wise comparisons reveal this differences primarily occur in the lower troposphere (0-2 km), likely due to ACTM's near-surface land $CO_2$ flux errors. In contrast, ACTM better matches aircraft $CO_2$ in the middle (2-5 km) and upper (5-8 km) troposphere, likely due to accurate large-scale transport representation. Over the Amazon, $CO_2$ differences with aircraft and OCO-2 differ, likely due to a lack of regional surface sites for inversion and insufficient high-altitude profile (~4km) not representative of XCO2. Over Asian megacity airports, which are significant emission hotspots, the model shows a large negative difference with CONTRAIL than OCO-2. This discrepancy likely hints that MIROC4-ACTM is unable to capture urban fossil $CO_2$ emission signals at airports due to coarse resolution (~2.8° x 2.8°) and higher resolution of OCO-2 limits ability to fully capture actual emission footprints.

**Keywords:** Carbon dioxide; Aircraft; Transport Model; OCO-2



**1. Introduction**

Atmospheric $CO_2$ is the most significant anthropogenic greenhouse gas (GHG) present in the Earth's atmosphere, responsible for a major global warming and climate change since the preindustrial era, circa 1750 (Canadell et al., 2022). Therefore, recognizing its importance in direct impact on climate, monitoring of highly accurate surface $CO_2$ measurements were first started from the South Pole (SPO) and Mauna Loa (MLO), Hawaii (Keeling, 1960) and later expanded across the globe. These in-situ measurements are widely used for estimating surface $CO_2$ fluxes using Bayesian-based "top-down" chemistry transport models due to their long-term record and high measurement accuracy (Chandra et al., 2022; Chevallier et al., 2010; Peylin et al., 2013). However, in-situ $CO_2$ measurement sites around globe is sparse, mostly situated in mid-latitude north America and Europe, with less coverage over tropical land (Patra et al., 2011; Schimel et al., 2015) and open oceans, which increase difficulties in inferring surface $CO_2$ fluxes from inverse models in data void regions (Chevallier et al., 2010, 2011).

To increase spatiotemporal monitoring of $CO_2$, spaced-based measurements such as SCanning Imaging Absorption spectroMeter for Atmospheric CartograpHY (SCIAMACHY), Greenhouse Gases Observing Satellite "IBUKI" (GOSAT), and the Orbiting Carbon Observatory-2 (OCO-2) were launched to provide column-average dry-air mole fraction or mixing ratio of $CO_2$, termed $XCO_2$ (Bovensmann et al., 1999; Crisp, 2008; Kuze et al., 2009). NASA's OCO-2 satellite launched in 2014 to achieve finer spatial resolution and better precision as compared to previous satellites. This advancement has proved beneficial for understanding global and regional carbon cycle science in various satellite based studies (Crisp, 2015; Das et al., 2023; Liang et al., 2017; Liu et al., 2017; Chatterjee et al., 2027). OCO-2 version 10 $XCO_2$ measurements has shown retrieval error with mean bias (RMSE) of 0.24 (0.81) ppm over land and 0.43 (0.84) ppm over ocean globally, compared against more accurate WMO scale maintained $XCO_2$ from surface-based Total Carbon Column Observation Network (TCCON) sites (Taylor et al., 2023; Wunch et al., 2017). Studies reported that assimilation of OCO-2 $XCO_2$ available at greater spatial density (~ 100 times of GOSAT) into an inversion requires the data to be extremely precise, stable and regionally unbiased to effectively estimate surface $CO_2$ fluxes in regional scale (Byrne et al., 2017, 2023; Crowell et al., 2019; Philip et al., 2022; Rastogi et al., 2021). Also, Miller et al. (2007) reported that satellite-retrieved $XCO_2$ needs regional precision of 1-2 ppm to reduce uncertainty in inversion-derived flux estimates from in-situ networks. Because, $XCO_2$ retrievals having many sources of uncertainty hinder their fidelity to utilize inversion approach to accurately estimate surface $CO_2$ flux (Chevallier et al., 2014; Villalobos et al., 2020). These retrieval errors in OCO-2 include cloud effects (Massie et al., 2021;Merrelli et al., 2015), instrumental errors, retrieved surface pressure, and then aerosol, the largest source of systematic error can be approximately 2 ppm over land regions (Connor et al., 2016). Therefore, to enhance the accuracy of surface $CO_2$ flux estimations, studies are focussing on improving retrieval algorithm by correcting for cloud effects and incorporating a digital elevation model (DEM) to correct surface pressure (Jacobs et al., 2023; Mauceri et al., 2023). Apart from retrieval errors, misrepresentation of transport and uncertainty in prior fluxes can further reduce reliability in top-down model inferred surface $CO_2$ fluxes (Chandra et al., 2022; Fu et al., 2021; Schuh et al., 2019). To address and assess such kind of errors impact on top-down $CO_2$ budgets, OCO-2 model intercomparison project (MIP) is formed with different $CO_2$ inverse modelling groups assimilating OCO-2, in-situ and combination of both ( https://gml.noaa.gov/ccgg/OCO2_v10mip/index.php).



Previous studies have attempted to reconstruct observation based $CO_2$ profiles combining ship, aircraft and model simulation to compare with GOSAT $XCO_2$, but are limited to characterise inversion errors (Müller et al., 2021; Inoue et al., 2013; Wofsy, 2011). Frankenberg et al. (2016), showed using HIPPO aircraft $CO_2$ vertical profiles, that the retrievals of GOSAT, TES, AIRS satellites and inversion simulation can have large difference of ~ 4 ppm due to inaccurate vertical transport in higher latitude during vegetation growing or decaying periods. These studies often lack in providing uncertainties linked with inversion through altitude-based $CO_2$ comparison from near surface to different tropospheric layers between inversion-based model simulations, surface, and aircraft data. This gap is particularly evident in regions with in-situ sparse data coverage, such as vast oceanic areas, as well as in emission or sink hotspots over land while OCO-2 dense measurements have not helped    to overcome precision issues providing global coverage. However, no studies attempted to understand how OCO-2 retrieval errors or accuracy hampers its full potential to uncover the uncertainties associated with inverse models. The present study aims to understand sources of error associated with the MIROC4-ACTM model through altitude-based $CO_2$ comparison among  MIROC4-ACTM, OCO-2 and aircraft observation across different tropospheric layers. To accomplish this we leverage highly accurate and precise aircraft vertical $CO_2$ measurements from ATom campaigns over northern America, Pacific, Atlantic, Southern Ocean regions, CONTRAIL over airports in Asia and four specific sites in Amazon. Before analysing, we first validated the MIROC4-ACTM model simulated tropospheric column $CO_2$ ($XCO_2$) with highly precise $CO_2$ vertical profiles from independent aircraft measurements     over open oceanic regions, Amazon  and local urban hotspot over Asia.

**2. Data and Methodology**

**2.1 Aircraft and surface $CO_2$ measurements**

We have used individual independent aircraft and surface in-situ $CO_2$ measurements around the globes from NOAA's obspack_co2_1_GLOBALVIEWplus_v8.0 data product (Schuldt et al., 2022) and WDCGG (World Data Centre for Greenhouse Gases) respectively. We have selected a few individual campaigns such as ATom, Amazon aircraft campaigns, CONTRAIL because of their extensive latitudinal/longitudinal spatial coverage over ocean and land regions, multiple vertical $CO_2$ profile measurements and extended period of measurements. A brief description of each aircraft measurement is discussed in the next paragraphs.

ATom is an aircraft field campaign, providing airborne measurements of remote tropospheric and lower stratospheric $CO_2$ from Northern America, Arctic, Pacific, Southern and Atlantic Oceans (Thompson et al., 2022). ATom has four campaigns conducted using NASA DC-8 aircraft, taking vertical profile of $CO_2$ from near surface (0.15 km) to13 km altitude range over four seasons from 2016 to 2018. Four campaigns each lasted around 28 days, namely, ATom–1, 28 July-22 August 2016; ATom–2, 26 January–22 February 2017; ATom–3, 28 September–26 October 2017; ATom–4, 24 April to 21 May 2018 respectively (Wofsy et al., 2021). These vertical $CO_2$ measurements enable the validation of $XCO_2$ measurements from the satellites because most of $XCO_2$ variability is constrained in the troposphere, therefore, these vertical measurements effectively serve as a reference for satellite-retrieved $CO_2$ validation (Frankenberg et al., 2016). We have utilised these $CO_2$ measurements freely available at https://gml.noaa.gov/ccgg/obspack/data.php.



Additionally, aircraft vertical campaign $CO_2$ measurements from four sites such as TEF (3.39° S, 65.6° W), SAN
(2.86° S, 54.95° W), RBA (9.38° S, 67.62° W) and ALF (8.80° S, 56.75° W) in the Amazon region are also
considered for the present study and more details about these measurements can be found in Gatti et al. (2021a).
In ALF, RBA, SAN, $CO_2$ measurements are available from 2010 to 2018, whereas TEF has measurements from
2013-2018. Bi-monthly vertical $CO_2$ profile measurements were taken during 12:00 and 13:00 local time at all
these sites covering altitude from 0.3 to 4 km when the daytime boundary layer is well mixed. These measurements
were taken using an automatic sampler onboarded in the light aircraft, which underwent accuracy and precision
testing at greenhouse gas laboratory at National Institute of Space Research (LaGEE/INPE), Brazil (Gatti et al.
2014). We have used a set of vertical $CO_2$ profiles    during September, 2014 till 2018, freely available at:
https://doi.pangaea.de/10.1594/PANGAEA.926834 (Gatti et al., 2021b).

Further, CONTRAIL aircraft program $CO_2$ measurements over Asian regions are also considered for the current
analysis (Ishijima et al., 2021; Machida et al., 2008; Matsueda et al., 2008). In this program, several regular
passenger aircraft operated by Japan Airlines (JAL) are installed with instruments like CME (Continuous $CO_2$
Measuring Equipment) to provide extensive spatial $CO_2$ data coverage in the upper troposphere and lower
stratosphere (UT/LS) region. We have only considered measurements during OCO-2 measurements period at four
representative zones in Asia, specifically around airports, to retrieve vertical $CO_2$ profiles during ascent or descent
of the aircraft, following Niwa et al., 2011. The measurements dataset is freely accessible at
https://www.cger.nies.go.jp/contrail/.

**2.2 OCO-2**

OCO-2 is a sun-synchronous satellite, retrieves $XCO_2$ to understand the carbon source-sink activity throughout
the globe (Eldering et al., 2017). The satellite uses three high-resolution grating spectrometers to retrieve the
reflected sunlight spectral signature of weak $CO_2$ (1.61 μm), strong $CO_2$ (2.06 μm) and $O_2A$ (0.76 μm) which are
later analysed through Atmospheric Carbon Observations from Space (ACOS) algorithm to estimate global
spatiotemporal $XCO_2$ distribution (Crisp, 2015; Crisp et al., 2017; Eldering et al., 2017). It has a spatial resolution
of 1.29 km × 2.25 km (nadir mode), and a temporal periodicity of sixteen days. We have utilised OCO-2 version
10 which is an update from previous version v8/v9 in terms of important changes in spectroscopy, aerosol, $CO_2$
prior source, and solar continuum model which reduced RMSE validated against $XCO_2$ measured at TCCON sites
for land and ocean-glint measurement (Taylor et al., 2023). In this study, we have used level–2 OCO-2 version
10r    data    product    (available    at:
https://disc.gsfc.nasa.gov/datasets/OCO2_L2_Standard_10r/summary?keywords=OCO-2) and have considered
only bias-corrected quality checked soundings ($XCO_2$_quality_flag = 0 or good data) for analysis.

**2.3 Transport model**

The  Model for Interdisciplinary Research on Climate, version 4, based Atmospheric Chemistry Transport Model
(MIROC4–ACTM) chemistry transport model is used, which is run at T42 spectral spatial resolution (~2.8° ×
2.8° latitude-longitude grid) with 67 vertical hybrid-pressure levels from the surface to 90 km to simulate the $CO_2$





concentration and fluxes (Patra et al., 2018). For realistic representation of the transport in the model, model
meteorology, horizontal winds (zonal and meridional) and temperature is nudged to Japanese 55-year Reanalysis
or JRA-55 data (Kobayashi et al., 2015). The MIROC4-ACTM model conducted two distinct simulations: one
utilizing bottom-up model a prior or "FG" fluxes and the other employing a posteriori or "InvFG" fluxes derived
from top-down inversion using 50 inversion sites (Figure S1). To derive total concentration in the simulation
different bottom-up tracers are utilised, gridded GridFED fossil fuel; $CO_{2,ff}$ (Jones et al., 2021), annually balanced
CASA biospheric flux; $CO_{2,lnd}$ (Randerson et al., 1997) and ocean exchange i.e., sea-air $CO_2$ flux; $CO_{2,ocn}$
(Takahashi et al., 2009). Then, prior $CO_2$ simulation case of MIROC4-ACTM is prepared by adding the prior
tracers as follows: $CO_{2,ff}$ (GridFED) + $CO_{2,lnd}$ (CASA–3hr) + $CO_{2,ocn}$ (Taka–Ocn). A detailed discussion on this
is given in Chandra et al. 2022. Further, to minimise the edge effect on the simulated dataset, we discarded the
first two years and last one year of our simulation period (2012-2022), only analysed $CO_2$ of both simulation and
observation during 2014-2021. Model performance evaluated by comparing with each vertical profile of
independent observations of ATom $CO_2$ not used in the inversion as well as at two surface sites, MLO (19.53 °N,
155.57 °W) and SYO (69.01 °S, 35.59 °W) representative of northern and southern hemisphere $CO_2$ variability
(Fig. S2, 3). To do that firstly, model simulated $CO_2$ is resampled to the nearest grid of the aircraft and surface
sampling locations, considering linear interpolations at spatial grid and time. We have not considered any other
co-location criteria unless it is mentioned, e.g.,     geometric and dynamic for comparison;     therefore, estimated
$CO_2$ difference is essentially uncertainty either in observation or inversion (Kulawik et al., 2016, 2019). The
result shows good performance with lesser bias with InvFG over prior at different latitudes, showing an overall
good match of InvFG $CO_2$ and ATom at different latitudes (Fig. S2, 3). Similarly at surface sites, InvFG showed
better performance over prior with correlation of 0.99 with observation (p < 0.05) (Fig. S4). Then, to compare
OCO-2 $XCO_2$ and ACTM-$XCO_2$, we formulated $XCO_2^{ACTM}$ following the Patra et al., 2017. Here, ACTM
simulated $CO_2$ profile or $CO_2^{ACTM}$ resampled at each OCO-2 retrieval location (latitude, longitude) with further
use of corresponding OCO-2 priori and column average kernel sensitivity (Ai) represents instrumental sensitivity
for 20 vertical levels from top of atmosphere (TOA) to surface to produce $XCO_2^{ACTM}$ using the following equation.

$$XCO_2^{ACTM} = \sum_i (CO_2^{priori} . dP_i) + \sum_i A_i . dP_i \left( \sum_i CO_2^{ACTM}_i - \sum_i CO_2^{priori} \right) \tag{1}$$
$CO_2^{priori}$ = OCO-2 priori;   $A_i$ = OCO-2 column averaging kernel;   $dP_i$ = thickness of each pressure layer.

We have also resampled $XCO_2^{ACTM}$ at 21 TCCON sites around the globe to validate model performance with
more accurate $XCO_2$ measurements at surface-based TCCON sites (Wunch et al., 2011). To perform that we first
filtered data points considering only good quality retrieval, then curve fitted the remaining retrieval to remove
outliers and finally considered retrievals with solar zenith angle < 60° following methodology mentioned in
Appendix C of Crowell et al. 2019. Figure S5 shows a good agreement between ACTM and TCCON $XCO_2$ for
majority of TCCON sites considering the fact that TCCON has its own bias due to topography, surface brightness
and aerosols as well as latitudinal varying bias (Wunch et al., 2017).

**2.4 Data analysis**



To conduct the spatial discrepancy analysis of $XCO_2$ difference between ACTM and OCO-2, bias corrected and
quality checked good soundings from OCO-2 $XCO_2$ retrieval or $XCO_2^{OCO-2}$ and ACTM simulated $XCO_2$ or
$XCO_2^{ACTM}$ is re-gridded into $0.5° \times 0.5°$ latitude-longitude grid boxes. Further, to assess this $XCO_2^{ACTM}$ -
$XCO_2^{OCO-2}$ difference, we used aircraft vertical $CO_2$ measurement available at different vertical tropospheric
layers, essentially to conduct an altitude-wise comparison of $CO_2$ among ACTM, OCO-2 and aircraft. Therefore,
we have employed a methodology outlined using a schematic in Figure 1, specifically designed to provide $CO_2$
profile from aircraft, OCO-2 and ACTM.
It shows typical $CO_2$ concentration vertical profiles and relative altitude range captured in OCO-2, aircraft
measurements, and corresponding MIROC4-ACTM simulations. OCO-2 measures $XCO_2$ concentration from
space, representing $CO_2$ profile from top of the atmosphere to surface so as the ACTM simulation at those pressure
levels is represented by a blue colour double-headed arrow. On the other hand, aircraft tropospheric columns of
$CO_2$ typically capture concentration variability up to an altitude of 15 km and ACTM resampled concentration
values at those measurement locations represented by orange colour double headed arrows. Since the main
purpose is to compare the aircraft tropospheric column $CO_2$ against OCO-2 $XCO_2$ and ACTM simulations, we
subdivided the tropospheric $CO_2$ column into three different vertical tropospheric layers, namely, the lower
troposphere: lowest level to 2 km, middle troposphere: 2–5 km, and upper troposphere: 5–8 km to understand the
model performance in each of the vertical layers. In addition, a total tropospheric vertical $CO_2$ column of aircraft
or aircraft $XCO_2$ is calculated only when the vertical measurements reach at least 8 km altitude; otherwise, any
profile not reaching 8 km is discarded from the analysis unless otherwise mentioned specifically. For each vertical
tropospheric layer, pressure-weighted partial column $CO_2$ is calculated to consider air mass variation between
pressure levels for each considered tropospheric layer. Moreover, to become more robust on the analysis, each
vertical depth layer is subdivided into 200 bins unless otherwise mentioned and 80% of vertical bins having
measurements for the specific tropospheric layer only considered for analysis. For instance, middle troposphere
(upper troposphere), i.e., 2–5 (5–8) km  is divided into 15 vertical bins each of 200 meters, then in a specific
latitude or longitude while calculating a partial column of $CO_2$, we only considered profiles that encompass a
minimum 12 vertical bins in them.

**3. Results**

**3.1. MIROC4-ACTM intercomparison with OCO-2 and surface measurements**

Figure 2a shows a monthly mean spatial $XCO_2$ difference  ($XCO_2^{ACTM}$ - $XCO_2^{OCO-2}$ ) during January, 2015 –
December, 2021. It reveals a heterogeneous signature of spatial $XCO_2$ difference or mismatch across the globe
with maximum negative $XCO_2$ difference of approximately 2 ppm over Amazon, Africa, south-east Asia, China,
primarily in the global tropical land regions.  This is likely due to the lack of long-term surface $CO_2$ measurements
(Figure S1) available for inversion particularly over the global tropics to constrain the prior $CO_2$ flux (also
discussed in Chandra et al., 2022). Further, humid tropics is also the region of lesser valid OCO-2 retrievals due
to the persistent shallow cumulus cloud blocking the infrared signals, makes it challenging to validate the transport
model and studies has shown sparse sampling over land increases chances of the error almost two times
(Frankenberg et al., 2024; Kulawik et al., 2019). On the other hand, high-latitude land regions, North America



and Russia, exhibit a positive $XCO_2$ difference of nearly 2 ppm, however, $XCO_2$ differences over ocean regions
are generally within ± 1 ppm, possibly because of lesser variability in ocean $CO_2$ flux compared to land $CO_2$ flux
and oceanic regions are minimally affected by land air mass. Further, we observed a negative $XCO_2$ difference
especially over northern extra-tropics between 30° N to 60° N of nearly -0.6 ppm in agreement with Byrne et al.
2023 likely attributed to OCO-2 ocean glint (OG) retrieval biases that adds up a layer of complexity of diagnosing
the transport model against OCO-2 retrievals over this vast ocean. Further, studies showed        that sampling
variance between land and ocean could also lead significant uncertainty (Basu et al., 2018). Additionally, we
observed a bias in the Southern Hemisphere, the underlying causes of which are still unknown and need further
research (Byrne et al., 2023). Overall, our results show some regions under/over-estimated by ACTM, however,
it is challenging to comprehend quantitatively and qualitatively about sources of error across diverse regions
(source-sink dynamics and transport mechanism) as it could result either due to inaccuracies in the inversion (prior
fluxes, transport) or errors in satellite retrievals (Chandra et al., 2022; Chevallier et al., 2014). Furthermore, to
check the time variation of these $XCO_2$ differences, we analysed the time versus latitude distribution of $XCO_2$
difference taking the average across global longitude from 180° W to 180° E (Fig. 2b). We observed that $XCO_2$
difference has a seasonal and spatially varying repeating signature, with maximum (minimum) difference during
February-March-April (September-October-November) consistent across study period. A prominent positive
(negative) systematic $XCO_2$ difference is observed over the southern hemisphere tropic to mid-latitude from 10°
S–40° S and northern latitude around 30° N (northern tropic to mid latitude) this is in agreement with Kulawik et
al. (2019). However, after separating time vs latitude distribution of $XCO_2$ difference into land and ocean, we
observed that this systematic $XCO_2$ difference mainly originates from the southern ocean part, which matches
well with overall (land and ocean) time vs latitude $XCO_2$ difference distribution (Fig. S6). However, it is
challenging to explain the difference since ocean glint has biases (Byr    ne et al., 2023) and a study by Kulawik
et al. (2019) also reported a systematic error of 0.6 ± 0.1 ppm could arise over the ocean and land in OCO-2
satellite $XCO_2$ retrievals.

To further examine whether this difference comes from inversion or $XCO_2$ retrieval because both have
uncertainties. We similarly analysed time versus latitude distribution of surface $CO_2$ concentration difference with
respect to ACTM simulation or $CO_2^{ACTM}$ - $CO_2^{In-situ}$ considering accurate surface $CO_2$ concentration data from 53
measuring sites around the globe. Most of these sites are situated in the northern hemisphere having at least 90%
data during study period (all sites geographical location can be visualised in Figure S7 in supporting information).
MIROC4-ACTM simulated $CO_2$ near the surface is resampled to the nearest grid of surface sites (latitude,
longitude, altitude) and measurement time from hourly interval model output. For each of the 53 sites, $CO_2$
concentration difference between ACTM and surface $CO_2$ concentration is calculated first, and then we linearly
interpolate it spatially as presented in Figure 2(c). Results show no such annually and spatially systematic
signature of $CO_2$ difference near the surface between equator to 45° S considering six stations (SEY: 4.7° S, ASC:
8° S, SMO: 14.2° S, NMB: 23.6° S, CPT: 34.4° S, CGO: 40.7° S) situated at different latitudes within this latitude
band (Fig. S7). Considering the fact that inferred $CO_2$ difference may arise due to much lower data density for the
in situ measurements within mentioned latitude bands, the analysis with the available sites suggests that systematic
signatures of difference exist when we compared with OCO-2 $XCO_2$ (Fig. 2b). It hints at uncertainties in OCO-2
retrieval or systematic vertical transport error in the model (Schuh et al., 2019), given the relatively lower





uncertainty in in-situ $CO_2$ measurements compared to OCO-2 $XCO_2$. This vast part of the region     remained
challenging for a model to understand its error characteristics due to OCO-2 retrieval error. Further, we also
compared the latitudinal average time series of $CO_2$ ($XCO_2$) difference of ACTM with surface (OCO-2) in Fig.
2(d). It shows an overall agreement of $CO_2$ difference with a correlation coefficient of 0.68 at 99% significance
level with total time series variability (1-σ STDEV) of 0.28 and 0.19 ppm in $CO_2$ difference in surface as compared
to OCO-2. Overall, this $CO_2$ space-time variability analysis clearly demonstrates that systematic signature in
$XCO_2$ difference, primarily concentrated in southern mid and northern high latitudes, previous studies also
indicated towards potential uncertainties may arise in retrieval over the ocean and or misrepresentation of vertical
transport in inversion (Byrne et al., 2023; Frankenberg et al., 2016; Schuh et al., 2019). Since systematic errors in
transport could result in inaccurate $CO_2$ flux estimates and, consequently, posterior simulated concentration (Deng
et al., 2015; Stephens et al., 2007). In an inversion estimation, transport, surface $CO_2$ flux and $CO_2$ spatial gradient
are closely interconnected so any misrepresentation in vertical or horizontal mixing consequently affects the
estimated flux. This highlights the complexity of interpreting $CO_2$ differences across diverse regions,
measurement platforms and error quantification of the optimised flux of inversions from surface and satellite
measurements. Therefore, to better understand the consistency of $CO_2$ differences across different global regions
and identify regions of major uncertainty which will enable us to address them effectively, we analysed $CO_2$
variation in different vertical tropospheric layers using vertical $CO_2$ profile datasets from aircraft measurements,
discussed in section 3.2.

**3.2 $CO_2$ difference in tropospheric layers**
**3.2.1 Over Globe**

Figure 3 represents mean $CO_2$ difference or $CO_2^{ACTM}$-$CO_2^{aircraft}$ across different latitudes using individual aircraft
observations for different tropospheric layers LT (light red), MT (orange), UT (dodger blue) and tropospheric
Total Column (teal), color coded to represent different altitude ranges. Aircraft measurements are generally
available in two modes: continuous measurements from the same site over a long period, and campaign
measurements that cover extensive vertical and horizontal distances with high data density over a limited period.
Therefore, we have subdivided aircraft measurements into two subcategories for our analysis: specific site
aircrafts having latitude coverage maximum 5°(Fig. 3a) and campaign aircrafts having latitude coverage
maximum 30° (Fig. 3b). Only those aircrafts having measurements during OCO-2 period are selected, each
aircraft sampling location, number of data points at different latitude bins of 30° and altitude bins of 1000 meters
is provided in the supplementary material (Fig. S8 and S9 gif for each aircraft category). Then, we calculated
model-observation $CO_2$ difference for each aircraft measurement category. Therefore, estimated $CO_2$ difference
serves as a model and observation mismatch for specific latitude (entire latitude range) for specific sites
(campaign) aircraft with latitude information mentioned inside parenthesis of first x-axis tick marks in Fig. 3a
(3b). Here, the second x-axis shows the number of data points in the corresponding aircraft campaign. Here, the
number of data points or samples is critical when comparing $CO_2$ differences among aircraft. A higher number of
samples provides better confidence to the calculated $CO_2$ difference while aircraft with fewer samples are
considered less weightage.



Mean (variability) of $CO_2$ difference for LT, MT, UT and tropospheric column for specific sites aircraft are -0.45
(± 0.49), -0.32 (± 0.48), -0.34 (± 0.5), and -0.2 (± 0.41) respectively (Fig. 3a). It shows the highest mismatch in
terms of mean exist in LT as compared to other tropospheric layers MT, UT and total tropospheric column, likely
due to uncertainty in prior $CO_2$ flux or transport in LT. Studies have shown that in the LT, concentration changes
are mainly regulated by surface $CO_2$ fluxes and diurnal-synoptic mixing patterns (Law et al., 2008; Patra et al.,
2008). However, $CO_2$ change in UT is mainly dominated by changes in large scale dynamical transport, where
surface emission has subdued influence. Hence, studies have found that coarse spatial resolution transport models
adequately simulated $CO_2$ in the MT to UT regions (Baier et al., 2020; Niwa et al., 2011). Additionally, it is also
noted that there is a systematic underestimation by the model in terms of magnitude in all tropospheric layers.
Result shows minimum or maximum (near zero) model-observation $CO_2$ difference (ppm) observed for "esp" or
"rta" (cma) aircraft at 49.48° N or 21.19° S (38.83° N) in the LT for specific sites aircraft. In the tropospheric
column, maximum (minimum) $CO_2$ difference is observed in "cma" ("car") aircraft at 38.83° N (40.66° N)
respectively. Further, it has been observed that the tropospheric column $CO_2$ difference matches pattern of $CO_2$
difference at LT at most aircraft sites; LT apparently contributes more to the total tropospheric column than MT
and UT. Further, it could be seen that the overall mean model-observation $CO_2$ difference is highest in the northern
mid-latitudes compared to the tropical latitudes of the northern and southern hemispheres.

Further, we have calculated $XCO_2^{ACTM}$ - $XCO_2^{OCO-2}$ at those specific sites aircraft location considering 5°× 5° grid
box surrounding it to check the difference similarity between $XCO_2$ and tropospheric column $CO_2$ selecting only
specific times from OCO-2 of aircraft measurement. Results show       $XCO_2$ difference mean (variability) of -
0.37 (±0.38), highlighting that the model is underestimating and also has minimum variability as compared to any
individual layers. On the other hand, $CO_2$ difference with campaign aircraft showed similar results; overall highest
mean and variability exists in LT observed. Further, the overall mean (variability) of the tropospheric column is -
0.39 (±0.1) controlled largely by $CO_2$ difference in LT. Here, we also check $XCO_2$ difference at those campaign
aircraft considering their covered tracks and then taking the average of all $XCO_2$ differences to calculate a mean
$XCO_2$ difference. Results show lesser variability in $XCO_2$ as compared to other layers and also negative mean
$XCO_2$ represents overall underestimation by the model. It has been observed in both aircraft categories that the
model has underestimated overall $CO_2$ concentration in all tropospheric layers and total columns, also the
maximum mean and variability of $CO_2$ difference are in LT. These differences are attributed possibly due to
underestimation by prior flux in the inversion or misrepresentation of transport in the model. To further investigate
$CO_2$ differences altitude-wise, we considered individual vertical $CO_2$ profiles from different campaigns for
different regions North America, Pacific, Southern Ocean, Atlantic, Amazon and Asia discussed in detail in
subsequent sections.

**3.2.2 North America, Pacific, Southern and Atlantic ocean**

Figure 4(a) illustrates integrated tracks traversed by the aircraft during ATom campaign (ATom-1, ATom-2,
ATom-3, ATom-4) across oceanic and land parts, subdivided into four segmented track categories corresponding
to specific geographical regions delineated with different colours. Represented segments are North America and
neighbours; east to west aircraft campaigning (magenta), Pacific; north to south aircraft campaigning (yellow),
Southern Ocean; west to east aircraft campaigning (red), and Atlantic; south to north aircraft campaigning (green).





Figure 4(b) shows the mean XCO$_2$ difference taken considering a collocation criteria of 5° × 5° latitude-longitude
grid box around sampling location and XCO$_2$ retrievals during corresponding ATom campaign period. We
observed a maximum XCO$_2$ difference of nearly 2 ppm over 120° W and 90° W in North America and
neighbouring land regions whereas oceanic regions, particularly Pacific and Atlantic are mostly confined within
±1 ppm at any specific latitude. We also checked the latitudinal bias MIROC4-ACTM against TCCON XCO$_2$
across the latitude during the ATom campaign period shown in Figure S11. This comparison also showed    higher
bias over this latitude location against TCCON sites at Park Falls, JPL, Lamont and East Trout Lake. Therefore,
model bias is consistent both against OCO-2 and TCCON. Then, to understand the altitude-wise variation of
CO$_2^{ACTM}$ - CO$_2^{aircraft}$ difference at different tropospheric depths (LT, MT, UT, tropospheric column), we used
ATom vertical CO$_2$ dataset. We also compared these CO$_2$ differences with OCO-2 across segmented ATom tracks
(Fig. 4c-f). Result shows largest CO$_2$ difference in terms of mean ± STDEV (calculated taking 1-σ  standard
deviation) of CO$_2$ differences across longitude range is -0.41 ± 0.94 ppm exist in LT as compared to the other
tropospheric layers likely due to uncertainty and large variability in prior land CO$_2$ flux near surface (Fig. 4c).
When we compare CO$_2$ differences from other layers of different track segments, North America and neighbour's
CO$_2$ difference at LT appears to be the highest, mainly occurring during the ATom-1 period (Figure S10).
Moreover, OCO-2 XCO$_2$ difference also showed large variability with longitudinal mean CO$_2$ difference of -0.34
± 1.07 ppm compared to aircraft tropospheric column CO$_2$ of -0.01 ± 0.48 ppm. This essentially reflects the
model's overall good performance against ATom as compared to OCO-2 XCO$_2$. Similarly, ACTM and ATom
CO$_2$ discrepancy was also evident in vertical cross-section, highest approximately ~ 2 ppm appeared at high
latitude land regions during vegetation growing (respiration) period of northern hemisphere July-August, 2016 in
ATom-1 (April-May, 2018 in ATom-4) (Figure S10). This large difference occurs across the vertical altitude
range prominent above 8000 meters likely arises due to the coarse resolution of the ACTM model unable to
represent the vertical transport. Needs further research on improving convective transport parameterization in
forward model to improve vertical mixing (Patra et al., 2018). These results are also in line with the study
Frankenberg et al., 2016. We have also validated model simulation with TCCON measurements during the ATom
period, results also shown differences up to 1 ppm at the sites over northern America (Figure S11).

Other three segments of CO$_2^{ACTM}$ - CO$_2^{aircraft}$ difference are primarily focused over oceanic regions (Southern,
Pacific, Atlantic) where magnitude of ocean CO$_2$ flux variability is less as compared to land regions as much as
10 times less.  Land and ocean CO$_2$ flux variability over the longitude and latitude band around ATom tracks for
different campaigns is shown in Figure S12. Figure 4d shows that over the Southern Ocean, the model-observation
CO$_2$ difference is lowest within ±0.2 ppm for each vertical tropospheric layer and 0.06 ppm for the aircraft
tropospheric column, with minimal variability compared to other layers. When compared with other ATom
segments, the Southern Ocean shows the lowest CO$_2$ difference in both mean and variability. This is because the
aircraft sampling is at the background troposphere and is farthest from land having little influence from land CO$_2$
air mass. This reflects that the optimized MIROC4-ACTM model, considering 50 ground-based sites, simulates
fairly well the aircraft background concentrations, however, it is unable to match a similar level of reproducibility
for OCO-2 XCO$_2$. Further, the latitudinal CO$_2$ difference variability against the aircraft tropospheric column
(lowest 8 km) is 0.15, compared to 0.77 with OCO-2 XCO$_2$. In most latitudes, OCO-2 XCO$_2$ differences are larger
than aircraft CO$_2$, indicating likely retrieval errors in OCO-2 given the lower uncertainty in aircraft CO$_2$





measurements. This $XCO_2$ difference with OCO-2 over the southern ocean is mainly during the ATom-2 period
(Fig. S14); however large $XCO_2$ difference pixels near 50° W and 70° W are during ATom-3 (September-October
2017; Fig. S15). Previous studies reported that OCO-2 ocean glint retrieval over southern ocean has more residual
biases while comparing against individual measurements (TCCON, In-situ, OCO-2 land) in top-down inversion
(Byrne et al. 2023; O'Dell et al., 2018). In addition, this is also location of the southern hemisphere zone of
stratosphere-troposphere exchange (STE) vary greatly spatiotemporally due to significant vertical mixing which
strongly changes with season, less constrained by model transport, which may also result in an error in estimated
posterior concentration. Next, we analysed $CO_2$ differences over the southbound Pacific segments of the ATom
campaign (Fig. 4e), it shows latitudinal $CO_2$ difference mean (variability) is highest of about -0.16 (± 0.53)  ppm
in LT, compared to MT and UT. On the other hand, aircraft tropospheric columns showed a mean (variability)
difference of approximately -0.04 (± 0.38) ppm whereas OCO-2 $XCO_2$ with a value of  0.27 (± 0.42) ppm. This
shows although the mean is significantly different however the variability in both $CO_2$ differences is close to each
other, this is reflected in overall matching of both aircraft columns and OCO-2 $XCO_2$ (Fig. 4e). Lastly, we
analysed $CO_2$ difference over the Atlantic i.e., longest northbound part of ATom campaign shown in Figure 4(f).
Aircraft tropospheric column $CO_2$ difference of aircraft shows value of 0.03 ppm as compared to $XCO_2$ difference
with OCO-2 showing value of 0.26 ppm. Although, it has been observed that the latitudinal $CO_2$ difference in the
aircraft tropospheric column closely matches pattern in OCO-2 $XCO_2$. Another important point is that overall
$CO_2$ differences variability in the Atlantic is observed higher as compared to the pacific segment especially over
tropics within 30° S–30° N, more in-situ aircraft measurements are     required to better understand the underlying
error. Individual ATom campaigns results are presented in supplementary Figure S13-16.

### 423  3.2.3 Amazon


The climate-sensitive global tropic is a crucial part of the global carbon cycle due to the threats posed by climate
change, especially the Amazon region, which holds the largest above-ground biomass (AGB) pool of
approximately 123 ± PgC (Malhi et al., 2006; Santoro et al., 2010). Inversion based estimate showed Amazon
was a carbon source of 0.3 ± 0.2 PgC/yr in agreement with bottom-up calculation (Alden et al., 2016; Beienen et
al., 2015; Gatti et al., 2014, 2021a,c) during 2010-2019, though significant uncertainty remain. This is also the
region under-sampled by OCO-2 retrievals due to clouds and high spatial resolution satellite monitoring is  needed
in the future (Frankenberg et al., 2024). Prevalent uncertainty in flux estimation in modelling approach and low
sampling of satellites highlight the need for more research in understanding error better way and improving both
inversion and retrievals methods over Amazon. In section 3.1, we observed large model-observation $XCO_2$
differences exist over South America, especially over Amazon. To investigate it further, we have utilised the
vertical profile (VP) of $CO_2$ measurements from vertical aircraft campaigns across Brazilian Amazon sites, SAN,
ALF, RBA, and TEF presented in Figure 5(a). The aircraft measurement has an accuracy of ~0.03 ppm (Gatti et
al., 2023) and a detailed description of measurements can be found in Gatti et al., 2021a. Studies have shown that
although these VPs are taken up to an altitude range of 4 km, they provide important insights into $CO_2$ variability
near the surface (Gatti et al., 2023; Tejada et al., 2023). An important point to note here, since Amazon aircraft
campaigns measure VPs of $CO_2$ approximately at an altitude up to 4 km, therefore, we considered the tropospheric
column as 0 to 4 km and only those VPs having measurements at least 4 km are chosen for calculation. We kept



the same criteria for the data availability of  80% vertical bin filter for 500-meter bin resolution as mentioned
previously in the methodology section.

Before analysing the ACTM bias against the Amazon aircraft $CO_2$, we validated ACTM simulated $CO_2$  with the
aircraft $CO_2$ at these sites which are not used in the inversion, showing a good correlation (r) of ~0.8 at 95%
significance level shown in Figure S17. We also checked that model able to capture TCCON $XCO_2$ at Manaus,
Brazil with time series $XCO_2$ difference mean of 0.05 ppm, but the model shows higher differences (-1 ppm) with
OCO-2 $XCO_2$ in Figure 2. It hints, OCO-2 retrieval likely have error could arise from Amazon dense vegetation
cover, cloud cover-aerosols and high humid conditions which can block sunlight spectra, reduce the signal
strength, limit valid sampling and increase retrieval error (Frankenberg et al., 2024; Taylor et al., 2016; Yu et al,
2019). These retrieval challenges precludes robust understanding of inversion error across the broader Amazonian
region.  To further check this error altitude wise, a monthly mean time series of ACTM-aircraft $CO_2$ difference
considering all vertical profiles within a month is calculated for three vertical tropospheric layers, LT (lowest–2
km), MT (2–4 km) and tropospheric column (lowest-4 km) during OCO-2 measurement periods is presented in
Figure 5c-f. Figure 5c represents $CO_2$ difference at SAN aircraft campaign sites having data gaps from mid-2015
to early-2017 because of no measurements conducted during this period. Maximum model-observation
differences in terms of mean(variability) of 0.93($\pm$ 3.36) ppm observed in LT as compared to MT and tropospheric
column. This mismatch is comparable with previous study by Basso et al. 2023. Further, OCO-2 $XCO_2$ difference
showed overall negative mean of -0.83 ppm with variability of $\pm$ 1.04 ppm as compared to aircraft VPs profile
with aircraft tropospheric column shown better constrained having value of 0.76 ppm. Further, we analysed VPs
at the ALF site presented in Figure 5(d) shows overall that aircraft model-observation $CO_2$ difference matches
well with $XCO_2$. $CO_2$ differences at LT, MT, tropospheric column, $XCO_2$ shows mean (STDEV) are -0.9 ($\pm$ 4.24),
0.08 ($\pm$ 2.03), -0.13 ($\pm$ 2.48), -0.65 ($\pm$ 1.03) respectively. Basso et al. 2023 has shown that some of this difference
between inversion and aircraft $CO_2$ could be significantly improved (57% below 1.5 km and 49% above 3.5 km)
when using regional  aircraft $CO_2$ data in the inversions. . In RBA, $CO_2$ difference at LT, MT, tropospheric
column, $XCO_2$ shows mean (STDEV) are -0.61 ($\pm$ 4.33), -0.03 ($\pm$ 2.52), 0.27 ($\pm$ 2.95), -0.69 ($\pm$ 1.04) respectively
shown in Figure 5e. Therefore, it shows $CO_2$ difference with the aircraft tropospheric column (OCO-2 $XCO_2$) has
opposite signature; it represents ACTM  over (under) estimates considering the whole time window. In TEF, $CO_2$
difference at LT, MT, tropospheric column, $XCO_2$ shows mean (STDEV) are -0.4 ($\pm$ 4.29), 0.64 ($\pm$ 3.02), 0.19 ($\pm$
2.89), -1.37 ($\pm$ 0.99) respectively presented in Figure 5f. Except for SAN, at all other sites, we observed that the
ACTM matches in total column better with aircraft than OCO-2, and but this profile is still insufficient to match
with $XCO_2$ needs further high profile measurement over this location. It is worth noting that the large discrepancy
or bias in LT in RBA, TEF (SAN, ALF) during January-March (August-December) in west-central (south-east)
Amazon regions may potentially arise due to fire $CO_2$ emission is reported in Basso et al. 2023. Since our inverse
simulations using CASA biospheric flux lack observation-based biomass burning data, this could also affect the
overall simulated concentration as well.  We also checked monthly land $CO_2$ flux anomaly, calculated by taking
area mean around campaign sites within $5° \times 5°$ degree and then removing seasonal cycle from actual time series.
We noticed no such anomalous flux change during the anomalous $CO_2$ difference period, likely due to the coarse
resolution of the MIROC4-ACTM and also because no regional $CO_2$ data from Amazon is used in our inversion
which could potentially capture Amazon land $CO_2$ flux changes better way (Fig. 5b; Fig. S1). Basso et al. 2023





highlighted the importance of assimilating Amazon aircraft measurements in deriving regional land $CO_2$ flux. In
all Amazon aircraft sites, an increase in land $CO_2$ flux during 2015-16 was observed due to strong ENSO events
occurred during this period also reported in Das et al. (2022).

**3.2.4 Asia**
In Asia there are very few aircraft campaigns for $CO_2$ measurements compared to Northern America and Europe
(Crevoisiera et al., 2010; Xueref-Remy et al., 2011). Although, efforts have been made to measure $CO_2$ vertical
profile over monsoon-dominated Indian subcontinents for a shorter time period (Vogel et al., 2023). Therefore,
available long-term $CO_2$ measurements like CONTRAIL is very important to provide unprecedented insights into
long-term $CO_2$ variability in UT/LS and model evaluations over these regions (Bisht et al., 2021; Das et al., 2022;
Niwa et al., 2011). Therefore, we have utilised these measurements to compare and understand model-
observations $CO_2$ difference for OCO-2 and CONTRAIL aircraft in different regions across Asia. Figure 6a
depicts the spatial distribution of the CONTRAIL campaign $CO_2$ sampling location from January 2015-December,
2021, covering altitudes ranging up to ~12 to 14 km with topographic altitudes information (topography elevation
data is downloaded from https://www.ncei.noaa.gov/products/etopo-global-relief-model). Here, we have selected
four separate regions around airport locations delineated through deep green colors having $CO_2$ vertical profiles
resulting from aircraft ascent or descent near airports. The four regions are namely Far East Asia,  Southeast
China, northern Southeast Asia and Equatorial Southeast Asia, based on the locations of airports. In Far East Asia,
two airports are considered: Tokyo International Airport, Japan (site code: HND) (35.6° N, 139.8° E) and Narita
International Airport (site code: NRT) (35.8° N, 140.4° E) are considered together, named TYO (35.7° N, 140.8°
E); in Southeast China, Hong Kong International Airport (site code: HKG) (22.2° N, 113.6° E); in northern
Southeast Asia, Suvarnabhumi International Airport, Thailand (site code: BKK) (13.7° N, 100.7° E); and in
southern Southeast Asia, Singapore Changi International Airport, Singapore (site code: SIN) (1.4° N, 104.0° E),
all airports are marked with a small square box in Figure 6a ,b. During the aforementioned period, no vertical
sampling was performed over the Indian subcontinent and other two airports highlighted on map Incheon
International Airport (site code: ICN) and Shanghai Pudong International Airport  (site code: PVG), were not
considered due to less number of sampling dataset. Figure 6b presents mean model-OCO-2 $XCO_2$ differences over
sampling locations of CONTRAIL, showing mainly negative $CO_2$ difference ranging -0.5 to -1 ppm over boxed
airports location highlighting likely reason is underestimation overall fossil emission of urban $CO_2$ signature in
the model. There is very limited TCCON sites over city scale that validates OCO-2 $XCO_2$, however, Rißmann et
al.    (2022) using Munich Urban Carbon Column network (MUCCnet) $XCO_2$ across three sites over Germany
found out OCO-2 has a RMSE of 0.6 ppm in urban site. Since OCO-2 has retrieval error over city scale it makes
it challenging to discuss the sources of error could come from the model.
To understand this we analysed more robust CONTRAIL aircraft $CO_2$ (~0.2 ppm for each CONTRAIL data point),
figure 6c-f represents a time series of model-observation $CO_2$ differences over each airport for different vertical
depths of troposphere and $XCO_2$. Here, we have considered all $CO_2$ vertical profiles, selecting aircraft ascent and
descent flight modes over airports within a month and done a monthly average for 200-meter vertical bins to
calculate partial column $CO_2$ for aircraft and similarly for model simulations resampled at aircraft measurement
location considering methodology described in section 2.4. For OCO-2, we computed the mean over designated




airports to calculate model-observation difference for XCO$_2$ for the specific months. Results show in far east Asia,
TYO location CO$_2$ difference in LT, MT, UT, tropospheric column, XCO$_2$ shows mean of -2.0, -0.88, -0.73, -
1.02, and -0.3 respectively. In HKG airport, the number of samples in LT was very less therefore ignored in the
analysis however CO$_2$ difference in MT, UT, tropospheric column, XCO$_2$ shows values of -1.2, -0.99, -1.13 and
-0.02 respectively. For northern Southeast Asia, BKK airport CO$_2$ difference at LT,  MT, UT, tropospheric
column, XCO$_2$ shows mean (STDEV) of -2.71 (± 1.67), -0.83 (± 0.74), -0.6 (± 0.59), -1.06 (± 0.71), and -0.1 (±
0.71)  respectively.  Further, in equatorial Southeast Asia, SIN airport difference at LT, MT, UT, tropospheric
column, XCO$_2$ shows mean (STDEV) of -1.89 (± 1.26), -1.03 (± 0.54), -0.81 (± 0.59), -1.05 (± 0.56), and -0.25
(± 0.42) respectively.
Result indicates that in northern Southeast Asia and southern Southeast Asia, mean and variability of model-
observation CO$_2$ difference is higher in LT as compared to UT, MT and weighs more. In all regions, model-
observation difference for OCO-2 showed better constrained compared to aircraft measurements and readily
observable that it closely matches the tropospheric column pattern. A notable fact in all regions is that the total
time series mean of model-observation difference is negative for both aircraft (-1.02 to -1.13 ppm) and OCO-2 (-
0.02 to -0.3 ppm), which would imply an underestimation of model simulated CO$_2$. While the OCO-2 XCO$_2$ vs
MIROC4-ACTM differences are not statistically significant but the large and systematic CONTRAIL CO$_2$ vs
MIROC4-ACTM differences may suggest that actual emission footprints captured by satellite observations are
greater than the measurement resolution (~1.29 × 2.25 km$^2$ for OCO-2). It suggests OCO-2 capturing emissions
from broader urban areas than its nominal resolution, possibly either due to the well-mixed nature of CO$_2$ and
OCO-2 measuring total column or the spatial extent of urban footprint. Further, the large variability and significant
differences between the aircraft CO$_2$ column and XCO$_2$ are evident in all regions. This is likely attributable to the
selection of a specific box area, which surrounds airport locations situated in urban areas, one of the significant
sources for fossil CO$_2$ emission. This inference is discussed in earlier studies (Patra et al., 2011; Umezawa et al.
2020), wherein they reported an urban emission footprint in CONTRAIL aircraft measurements conducted over
airport megacities. The inversion process, utilized in this context, exclusively optimizes total CO$_2$ fluxes, for
biosphere and ocean regions considering background sites, whereas this CONTRAIL measurement over airports
having signature of urban interiors. Consequently, noteworthy disparities may emerge due to uncertainties
associated with fossil fuel CO$_2$ emissions and also coarse horizontal resolution of MIROC4-ACTM (T42, ~2.8°
× 2.8°) unable to reproduce the sub-grid-scale variations. This limitation does not influence optimized flux for the
large area studies but affect our ability to simulate posterior concentrations, leading to underprediction of
concentration near the surface over the emissions or sinks hotspots, e.g., anthropogenic emissions at the megacity
areas or plumes of intense biomass burning. Note that the location of ascent and descent of the aircraft may change
by seasons following the meteorological conditions, and thus the location of measurements less strictly follows
year around. Previous studies underscore the critical role of fossil fuels in shaping simulated CO$_2$ dynamics,
emphasizing their potential to introduce systematic errors in optimized surface fluxes (Suntharalingam et al.,
2005; Wang et al., 2020).
**3.3 Discussion and conclusions**



The availability of the OCO-2, aircraft, and in-situ $CO_2$ observations, along with MIROC4-ACTM simulation at
their corresponding measurement location and time, allows us with an opportunity to understand fine-scale $CO_2$
difference between the ACTM and OCO-2 more robustly. Because, this enables diagnosing $CO_2$ difference from
near surface to different tropospheric layers, utilising  surface, and aircraft observation to highlight persistent
limitations in addressing inversion uncertainties.
●   We demonstrated that MIROC4-ACTM, using only 50 surface-based $CO_2$ sites globally, accurately
simulates tropospheric column and OCO-2 $XCO_2$, showing strong agreement with aircraft and TCCON
data (correlation = 0.9, p < 0.0001) at most sampling sites.

●   Our analysis highlighted that the regional hemispheric MIROC4-ACTM $CO_2$ difference with OCO-2
and in-situ measurements has heterogeneous signatures of $CO_2$ differences, particularly  over  land.
However, Kulawik et al. 2019, noting that OCO-2  retrievals over lands have more random errors
especially over Amazon which is less sampled by OCO-2 makes retrieval less reliable for comparison
(Frankenberg et al., 2024). Additionally, comparison against in-situ indicates that OCO-2 likely has a
systematic retrieval error over the southern hemisphere oceanic region. Both random error over land, less
sampling of OCO-2 over global tropics and systematic error over ocean makes it difficult to detect and
understand the uncertainties in inverse models in a global perspective. We need more vertical aircraft
profile measurements to more robustly understand this error especially over the global tropics.

●   Altitude wise comparison of $CO_2$ difference from categorical specific and campaign aircraft
measurements around the globe consistently highlight the  model's highest mismatch in LT as compared
to MT, UT, and tropospheric columns. Additionally, LT contributes  more to the  mean and variability
to the total tropospheric column than the MT, UT. This maximum uncertainty in the LT likely arises
from the  uncertainties in prior fluxes near the surface. In contrast, the MT and UT, where large-scale
dynamical mixing predominates, show better model performance, likely due to realistic transport of the
forward model.      Further, aircraft tropospheric $CO_2$ columns are better reproduced by MIROC4-ACTM
compared to individual tropospheric layers and OCO-2 $XCO_2$. Further, studies have shown OCO-2
$XCO_2$ is more prone to erroneous retrieval due to near surface     aerosol and cloud contamination in LT
which makes it challenging for total column comparison with model (Connor et al., 2016; Massie et al.,
2021).

●   Results from ATom show large $CO_2$ difference variability over North America regions as compared to
other integrated tracks over ocean, likely because of the influence of land air mass having large variability
in land $CO_2$ flux. Similarly, large $CO_2$ differences in aircraft sites over Amazon may likely arise due to
uncertainty in prior flux and coarse resolution of the model unable to represent small scale variation
requires more regional measurements in inversion, however, comparison against OCO-2 highlights
robust requirement of good amount of valid retrievals to diagnose the inversion from large region
perspective as well as insufficient high-altitude profile measurements (~4 km) demands more high profile
measurements. Aircraft measurements over the remote background troposphere in the Pacific, Southern
Ocean, and Atlantic showed the best match within 0.03 ppm when compared to OCO-2  with 0.2 ppm,



especially over the Southern Ocean. However, the model comparison with CONTRAIL has shown
consistent more (less) underestimation against aircraft (OCO-2) $CO_2$ measurements for all airports in
Asia. This discrepancy is likely due to the coarse resolution of the inversions unable to capture the
signature of urban fossil $CO_2$ emissions and also for OCO-2 unable to capture the whole urban footprint.

**Code availability**
Data and figure processing codes prepared for this study will be made available upon reasonable request from the
corresponding author. Inversion code used here is available on https://github.com/prabirp/co2l2r84.
**Data availability**
The ObsPack data product is available at https://gml.noaa.gov/ccgg/obspack/, CONTRAIL
at https://www.cger.nies.go.jp/contrail/, Amazon aircraft campaign at
https://doi.pangaea.de/10.1594/PANGAEA.926834. ACTM model outputs at aircraft sampling and OCO-2
sounding locations will be made accessible upon requests from corresponding authors.
**Author contributions**
CD, PKP and RKK developed the idea and methodology of the study. CD ran the entire analysis and prepared the
main manuscript and supplementary. PKP, RKK and NC helped in review and editing. KI, and TM are part of the
CONTRAIL $CO_2$ measurement group. All co-authors actively engaged in scientific discussions.
**Competing interests**
The authors declare that they have no conflict of interest.
**Acknowledgments**
All co-authors sincerely thanks to National Oceanic and Atmospheric Administration (NOAA), Japan
Meteorological Agency (JMA), Commonwealth Scientific and Industrial Research Organisation (CSIRO),
Laboratoire des sciences du climat et de l'environnement/Institut Pierre Simon Laplace (LSCE/IPSL), Scripps
Institution of Oceanography (SIO), South African Weather Service (SAWS), Environment and Climate Change
Canada (ECCC), Mt. Waliguan/ China Meteorological Administration (WLG/CMA), Tae-ahn Peninsula/Korean
Meteorological Administration (TAP/KMA) for site observations, and the CONTRAIL, Amazon, ATom
campaign teams for the aircraft observations. Surface in-situ $CO_2$ observations from Jungfraujoch are supported
by the Swiss Federal Office for the Envrionment and ICOS Switzerland (ICOS-CH). This study would not have
been possible without these $CO_2$ concentration measurements datasets around the globe.

**Financial support**

CD thanks to the Director of IIT Delhi, for providing PhD fellowship to carry out this work. This research is partly
supported by the Environment Research and Technology Development Fund (JPMEERF24S12205) of the



Environmental Restoration and Conservation Agency of Japan, and the Arctic Challenge for Sustainability phase
II (ArCS-II; JPMXD1420318865) Projects of the Ministry of Education, Culture, Sports, Science and Technology
(MEXT).

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




**Figures**

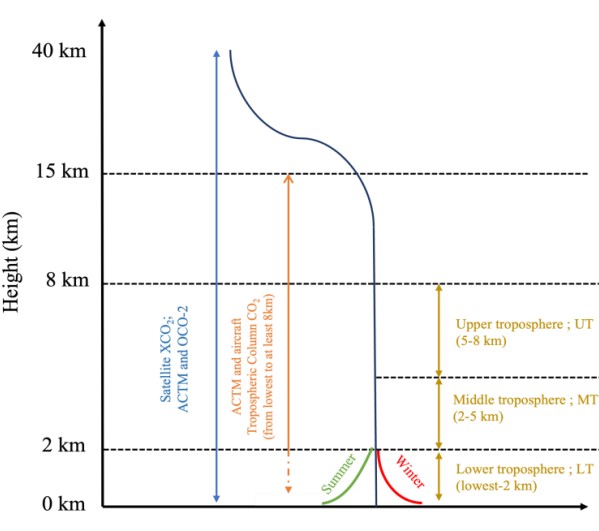


**Figure 1:** Schematic of $CO_2$ concentration vertical profile (dark blue line) by satellite, ACTM, and aircraft $CO_2$
(orange, golden) Arrowheads represent different layers of the atmosphere, specifically LT (lowest–2 km), MT
(2–5 km), UT (5–8 km), and the tropospheric column (lowest–8 km) corresponds to the aircraft $CO_2$
measurement. Blue arrow represents total variation captured by satellite covers from surface to top of the
atmosphere.

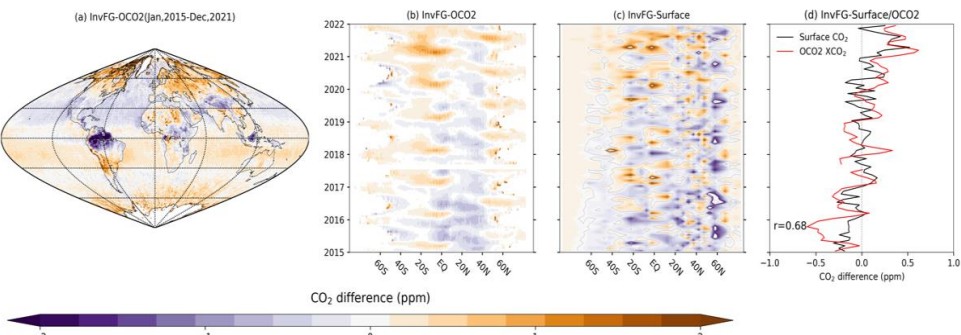


**Figure 2:** $CO_2$ space-time variability with (a) spatial mean $XCO_2$ difference map between InvFG and OCO-2
during January,2015-December,2021. (b) Time vs latitude distribution of $XCO_2$ difference between InvFG and
OCO-2 considering mean across global longitude. (c) Time vs latitude cross-section of $CO_2$ concentration
difference between InvFG and in-situ $CO_2$ measurement, considering $CO_2$ from 53 surface sites. (d) Latitude





averaged time series of $CO_2$ ($XCO_2$) concentration difference between InvFG and Surface (OCO-2) respectively
represented by black (red) colours. "r" value in panel-d represents correlation between time series of surface and
OCO-2 difference at 99% significance level.

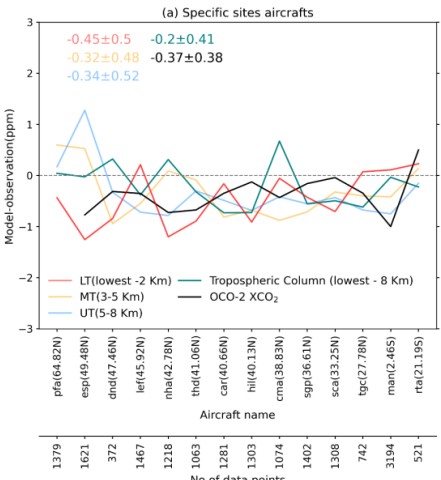

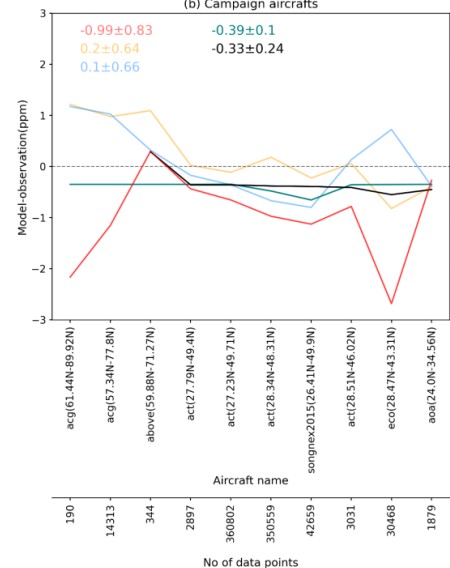


**Figure 3.** Mean model-observation $CO_2$ difference (ppm) at different vertical tropospheric depths LT (light red),
MT (orange), UT (dodger blue), total column (teal) and $XCO_2$ for specific sites aircraft (panel–a) and campaign
aircraft measurements having latitudinal coverage maximum 30° (panel–b). Aircrafts names are organized based
on aircraft observations location, progressing from high latitudes in the Northern Hemisphere, through the equator,
to southern latitudes. The second x-axis represents a number of data points for specific aircraft observation. The
first and second number inside the panel represents mean and 1-σ standard deviation (STDEV) of model-
observation difference across latitude for each tropospheric layer.

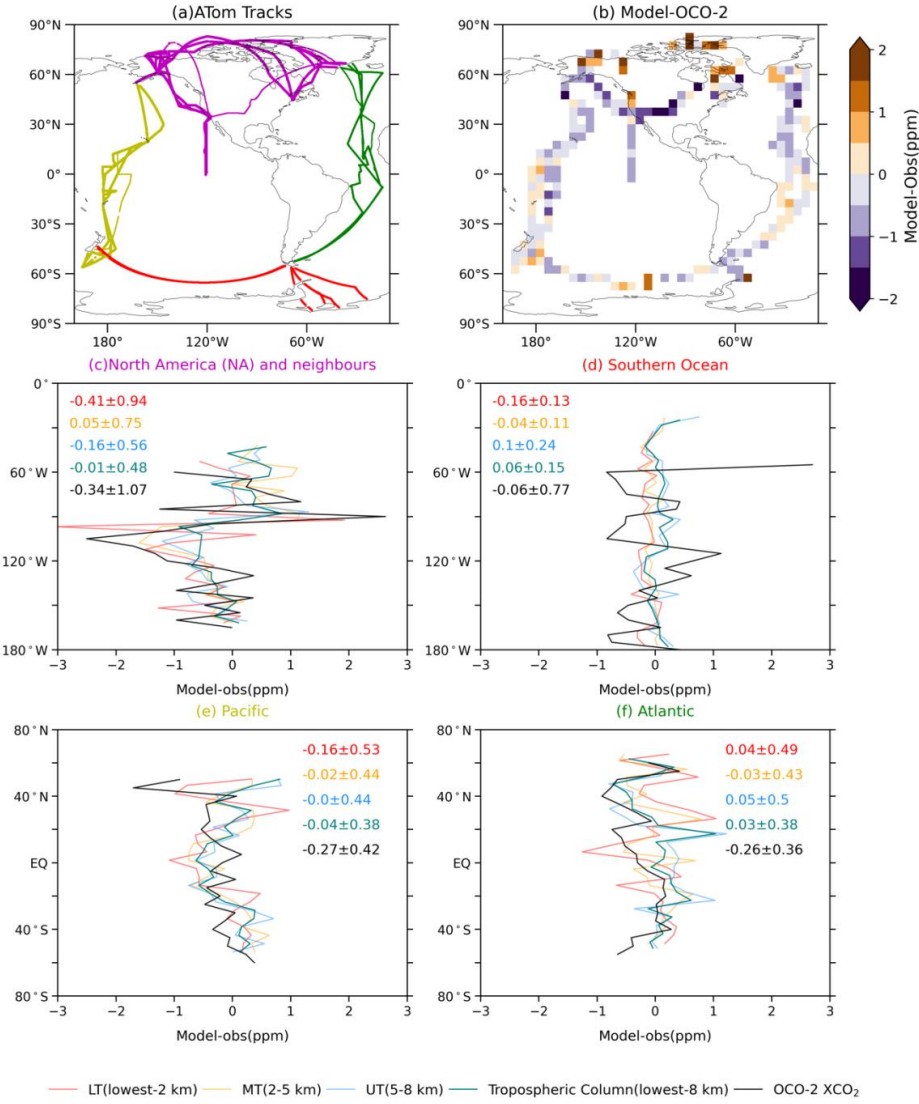

**Figure 4:** (a) Integrated tracks traversed during ATom campaigns (ATom-1, ATom-2, ATom-3 and ATom-4).

(b) Spatial model-observation $XCO_2$ difference against OCO-2 over ATom integrated track during campaign (c),

(d), (e), and (f) shows model-observation $CO_2$ difference over different tropospheric layers from vertical $CO_2$

profile measurements of ATom and $XCO_2$ from OCO-2 for North America and neighbours, Southern Ocean,

Pacific, and Atlantic segments respectively. Tropospheric layers are LT (light red), MT (orange), UT (dodger

blue) and Total Column (teal), and OCO-2 $XCO_2$ (black) representation for difference against OCO-2. The first

and second number on the right side of each middle and bottom panel represents the mean and 1-σ standard

deviation (STDEV) of model-observation difference across latitude or longitude, respectively.




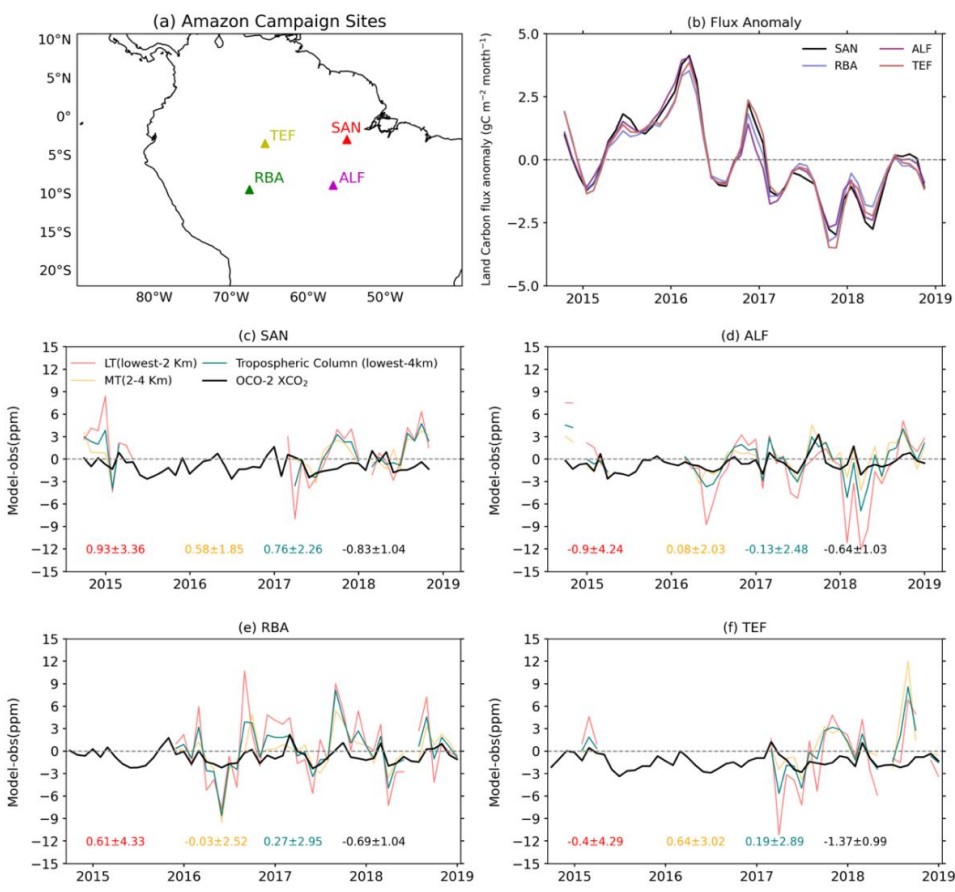

**Figure 5:** (a) Amazon aircraft vertical $CO_2$ profile campaign sites, SAN, ALF, RBA, and TEF. (b) Time series of
land carbon flux anomalies at the campaign sites. (c), (d), (e) and (f) represent a time series of model-observation
$CO_2$ differences for LT, MT, and tropospheric column, and $XCO_2$ during OCO-2 measurement periods for SAN,
ALF, RBA and TEF, respectively. The numbers inside the middle and bottom panels represent the mean and 1-σ
standard deviation (STDEV) of model-observation $CO_2$ difference over the time period.



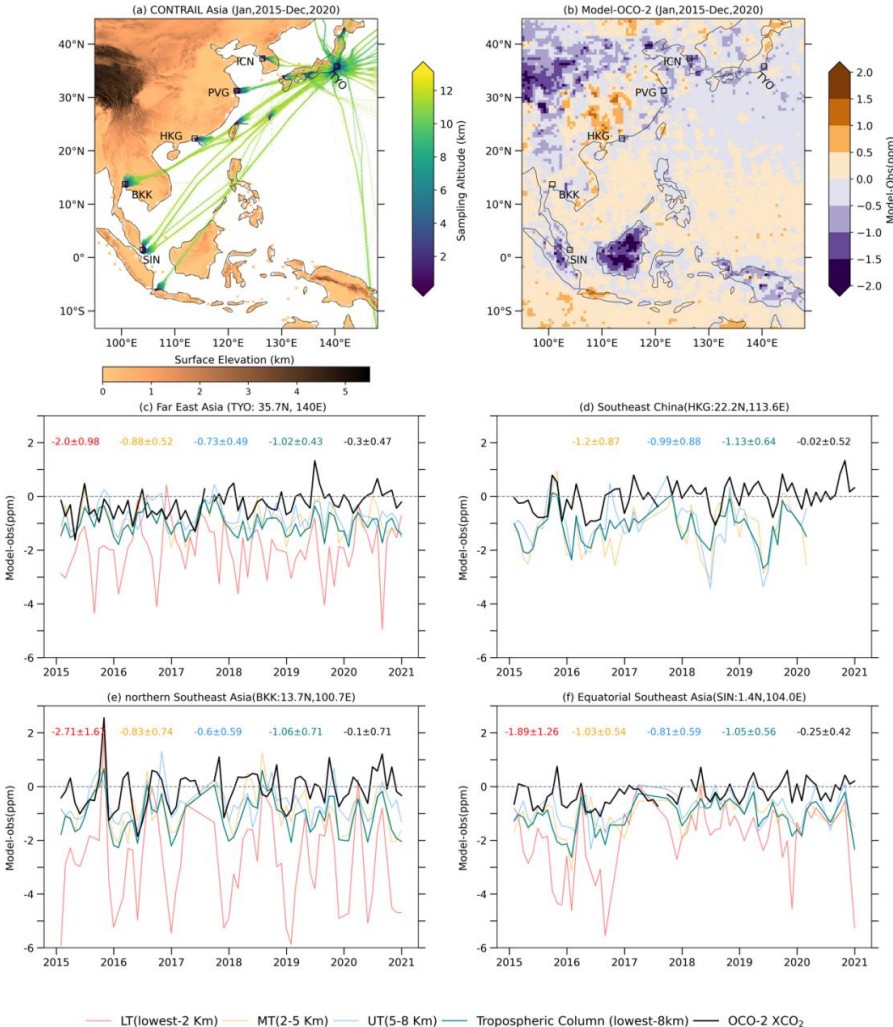

**Figure 6:** (a) CONTRAIL aircraft sampling locations with associated colours represent sampling altitude in km over Asia regions with surface elevation in km. There are four defined regions, each associated with specific airports, covering various vertical zones of carbon dioxide ($CO_2$) profiles. Far East Asia consisting of two airports, HND, NRT merged to prepare TYO, Southeast China with one airport HKG, northern Southeast Asia with one airport BKK, and equatorial Southeast Asia encompassing one airport SIN. (b) mean model-OCO2 $XCO_2$ differences over sampling locations during the mentioned period. (c), (d) (e), and (f) are time series of model-observation $CO_2$ differences over representative airports at different vertical depths of troposphere MT, LT and tropospheric column and $XCO_2$ utilising aircraft measurement and OCO-2. The numbers inside the (c), (d), (e), and (f) panels represent the mean and 1-σ standard deviation (STDEV) of model-observation difference over the period.