# Peer review of "Uncertainties in OCO-2 satellite retrievals of XCO2 limit diagnosis of 1 transport model simulation uncertainty 2"

_EGUsphere, 2024_

## Author Comment (AC1)

1. Das et al. analyzed differences between MIROC-ACTM simulated atmospheric $CO_2$ profiles and observations from aircraft and OCO-2. While the paper's title claims to focus on understanding how OCO-2 satellite retrieval uncertainties limit the diagnosis of transport model simulation uncertainty, the actual analysis presented does not address this question. The paper is poorly written with unclear reasoning and numerous grammatical errors. The quality does not satisfy ACP standards. I do not recommend this paper for publication based on the following major concerns:

[Reply] We sincerely thank the reviewer for the time and effort in reviewing our manuscript and highlighting the major concerns. We also appreciate the comment on grammatical errors. In the revised version, we have tried our best to correct the writing style, and we have also made extra analysis, statistical significance tests, updating figures, and summary tables. Changes are reflected in the abstract and in the other sections.

Because the revisions are very significant throughout the manuscript, a revised draft is supplied with this reply for your reference, which will be further improved for clarity when we are allowed to revise our discussion article.

2. Scope and Focus: Analyzing transport uncertainty requires examining tracer (e.g., $CO_2$) vertical and meridional profiles. However, satellite retrieval algorithms have known limitations in resolving vertical details from $XCO_2$ measurements. Using satellite vertical $CO_2$ retrievals to analyze transport uncertainty is fundamentally limited. A meaningful study should examine how biases in vertical profiles derived from OCO-2 $XCO_2$ could affect transport analysis (e.g., vertical mixing or PBL mixing). The manuscript fails to address these fundamental aspects of transport uncertainty analysis.

[Reply] We appreciate your comment. ACTM simulations have been evaluated against both vertical and meridional $CO_2$ profiles (~10-14 km for ATom) (Figure S2; https://zenodo.org/records/14504067). For better clarification, we modified L160 sentence to: "*Then, the model simulation is evaluated by comparing with zonal and meridional vertical $CO_2$ profile from ATom campaign as well as at two surface sites, MLO (19.53 °N, 155.57 °W) and SYO (69.01 °S, 35.59 °W), representative site of $CO_2$ variability in the northern and southern hemisphere (Fig. S2,3,4)*".

While it is true that satellite retrievals have limitations in resolving vertical details, however, sustained efforts have progressively reduced retrieval error (Kiel et al., 2019; O'Dell et al., 2018; Taylor et al., 2023). Despite these limitations, satellite retrieval remains essential tracking global $CO_2$ variation due to the sparsely situated in-situ sites worldwide. Therefore, we believe understanding retrieval uncertainties is critical for further improving their accuracy and transport model evaluations. Various revisions to the manuscript is made to show the usefulness of OCO-2 data.

3. Methodological Issues: Analysing model or data biases always requires establishing a 'ground truth'. This paper compares MIROC-ACTM simulated atmospheric CO2 profiles with observations from aircraft and OCO-2. However, it is not clear what is considered as the truth and what is being analysed. The authors keep switching between these three models/products to evaluate the other two. In some cases, the authors consider MIROC-ACTM model simulation as the truth, which has certain limitations. My understanding is that the model output is the forward transport of a posterior flux, which was derived using only surface station data. However, the column-averaged model data might still be biased due to transport errors of the parent transport model and any additional bias from the inversion setup. It is fine to establish this model output as the truth, but the authors need to show that the column average could represent the observed column average.

[Reply] We would like to clarify that we have not considered the ACTM output as the 'ground truth'. We used surface-based and aircraft $CO_2$ measurements as truth, as they provide the most accurate $CO_2$ measurements. To avoid any confusion we have clearly mentioned it in the abstract "*In this study, we used MIROC4–ACTM simulations, surface and aircraft observations (ATom, Amazon, and CONTRAIL projects - considered as ground truth), and Orbiting Carbon Observatory–2 (OCO–2) $XCO_2$ covering*

*2015–2021*". We have also mentioned in introduction(L83) and data and methodology section (L100-102).

To ensure a rigorous ACTM evaluation vertically, we validated ACTM simulations against vertical $CO_2$ profile (~12-14 km) from ATom campaign (Fig.S2; https://zenodo.org/records/14504067) and column average ($XCO_2$) (top of the atmosphere to surface) from TCCON sites (Fig. S5). This is mentioned in the data and methodology section.

4.The method description part lacks many details. For example, how was the column average calculated? How do you compare $XCO_2$ with aircraft column averages, considering they have different spatial coverage? Overall, the sampling strategy comparisons between different observation types need more careful consideration. The criteria for comparing different observational datasets need better justification. The statistical treatment of uncertainties requires more rigour. For example, what do you mean by variability? When you say there is a difference, is it significant?

[Reply] Great Point! We added the following sentence in the data analysis section (L220), "*Then, pressure-weighted partial column $CO_2$ is calculated for each of this tropospheric layers considering air mass variation between pressure levels (Chandra et al., 2017) to compare with OCO–2 $XCO_2$ and assess ACTM simulation*".

We appreciate your question. We would like to clarify first that ACTM simulated $CO_2$ is sampled at the nearest grid of aircraft, surface, and OCO-2 observation locations and closest time of measurements within 0.5 hour. Following the sampling, we take spatial and time averaging appropriately before presentation of model-observations.

**Comparisons of MIROC4-ACTM simulation with OCO–2 retrievals and surface measurements:** Here, we calculate the monthly differences in $XCO_2$ between ACTM and OCO-2. Following a similar approach, we also compute the $CO_2$ differences between ACTM and surface observations for the corresponding months. This is clearly mentioned in section 3.1 (L232, L275).

**Comparisons of MIROC4-ACTM simulation with OCO–2 retrievals and aircraft measurements:** We considered OCO–2 retrievals of $XCO_2$ within a 5° × 5° grid centered around the aircraft sampling locations and for the same dates. This is mentioned on L324-326. We have revised the text to convey this clearly without confusion and it reads as follows: "*We also calculated differences in $XCO_2$ between the model and OCO–2 by considering $XCO_2$ data points within a 5°×5° grid box centered around aircraft sampling locations and during aircraft $CO_2$ measuring dates (co-located aircraft and satellite)*".

"*Variability*" refers to the one-sigma standard deviation (1σ-STDEV) of the $CO_2$ differences between model simulations and observations. This variability is calculated either across time (e.g., for Amazon and CONTRAIL campaigns) or across space (e.g., for ATom campaign). We have clearly mentioned this usage at relevant parts and also captioned (L338, L363-365, L442, etc) it in the figure in the revised manuscript.

Thanks for raising this point. Following your suggestion, we have done one-sample t-test for checking the statistical significance of the estimated $CO_2$ difference. Now, p-values are included along with $CO_2$ differences values in the revised draft. We added the following sentence on L226 in the data analysis section, "*Lastly, to assess the significance of $CO_2$ differences in each vertical layer, we applied a one-sample t-test (Gerald, 2018).*"

5. Writing and Organization: The manuscript suffers from severe organisational and writing issues that significantly impair its readability and scientific credibility. The logical flow is persistently problematic, with paragraphs lacking clear structure and transitions, making it extremely difficult to follow the authors' reasoning. Most sections and paragraphs lack topic sentences to guide readers through the

argument progression. The paper's overall organization is confusing, with little logical connection between sections, making it challenging to understand how different components of the analysis relate to each other. The manuscript is also filled with grammatical errors, suggesting it has not been properly proofread. The writing style lacks scientific rigor and precision - many statements are made without proper justification or clear explanation, and technical terms are used inconsistently. Here are a few examples (I cannot list all instances due to the extensive number of vague presentations and grammatical errors).

[Reply] We have reorganized the draft by restructuring sections into clear, concise paragraphs with smooth transitions, consistent terminology, and focused topic sentences with proper justifications (to the best of our ability). Additional improvements include corrected grammar, new analyses, statistical significance tests, updated figures with data counts, and a new summary table. We sincerely hope that the article now qualifies the standard of ACP.

1. L242: "lesser" is incorrect usage

[Reply] Thank you very much for this and other minor suggestions below. We have used "less" instead.

2. L244: "Negative difference" is vague; better to specify low or high

[Reply] Done. Now "Negative difference" is replaced with "low" here and other places.

3. L249: The bias referenced is undefined

[Reply] We removed the sentence. We find it unnecessary.

4. L250: "some regions" is too vague

[Reply] We find it vague and unnecessary so we have removed it in the revised version.

5.L269: "CO2_in-situ" is vague, as aircraft data could also be in-situ measurements

[Reply] Thanks for spotting that. We have changed it to: "CO2_Surface" to avoid the confusion.

6.L278-280: The focus of this sentence is unclear

[Reply] Deleted this sentence since we find this statement redundant here.

7.L282: "The vast part of the region" - which region?

[Reply] To avoid misinterpretation, we revised the sentence to:" *Since surface-based $CO_2$ measuring sites are only available at specific locations, it remains challenging to evaluate model near the surface in region without $CO_2$ monitoring stations*"

8.L349-350: "These differences are..." - this statement lacks supporting analysis

[Reply] We have added citation and the sentence now reads as "*This high difference in LT is likely due to uncertainties in prior surface $CO_2$ flux, poorly constrained by the sparse surface measurement network of 50 sites, as previous studies have shown that $CO_2$ change in the lower-most model layer is predominantly influenced by surface $CO_2$ fluxes (Law et al., 2008; Patra et al., 2008)*".

9.L378-379: "This essentially..." - this statement lacks supporting evidence

[Reply] We deleted this sentence and added the following sentence to bring more clarity "*A high difference, however, is observed near 60° W, which coincides with a region of very few OCO–2 XCO$_2$ data points around 60° S,60° W (Fig. S13–ATom-3)*"

10.L384: "Needs further …" is an incomplete sentence

[Reply] We have revised the sentence to improve readability "*Dynamical process in this region, related to the stratosphere-troposphere exchange, can lead to steep CO$_2$ gradient between upper troposphere and lower stratosphere during the austral spring and autumn season i.e., ATom-3 and ATom-4 period respectively. This gradient is less constrained in ACTM transport may also contribute to this observed difference (Fig. S10; Bisht et al., 2021)*"

11.L425-429: This content belongs in the introduction/motivation section

[Reply] We agree with the reviewer's comment. We have deleted these sentences, as we also found them unnecessary at this point.

**References:**

Chandra, N., Patra, P. K., Niwa, Y., Ito, A., Iida, Y., Goto, D., Morimoto, S., Kondo, M., Takigawa, M., Hajima, T., and Watanabe, M.: Estimated regional CO$_2$ flux and uncertainty based on an ensemble of atmospheric CO2 inversions, Atmos. Chem. Phys., 22, 9215–9243, https://doi.org/10.5194/acp-22-9215-2022, 2022.

Kiel, M., O'Dell, C. W., Fisher, B., Eldering, A., Nassar, R., MacDonald, C. G., and Wennberg, P. O.: How bias correction goes wrong: measurement of XCO2 affected by erroneous surface pressure estimates, Atmos. Meas. Tech., 12, 2241–2259, https://doi.org/10.5194/amt-12-2241-2019, 2019.

Law, R. M., Peters, W., Rödenbeck, C., Aulagnier, C., Baker, I., Bergmann, D. J., Bousquet, P., Brandt, J., Bruhwiler, L., Cameron-Smith, P. J., Christensen, J. H., Delage, F., Denning, A. S., Fan, S., Geels, C., Houweling, S., Imasu, R., Karstens, U., Kawa, S. R., … Zhu, Z.: TransCom model simulations of hourly atmospheric CO2: Experimental overview and diurnal cycle results for 2002. Global Biogeochemical Cycles, 22(3). https://doi.org/10.1029/2007GB003050, 2008.

O'Dell, C. W., Eldering, A., Wennberg, P. O., Crisp, D., Gunson, M. R., Fisher, B., Frankenberg, C., Kiel, M., Lindqvist, H., Mandrake, L., Merrelli, A., Natraj, V., Nelson, R. R., Osterman, G. B., Payne, V. H., Taylor, T. E., Wunch, D., Drouin, B. J., Oyafuso, F., Chang, A., McDuffie, J., Smyth, M., Baker, D. F., Basu, S., Chevallier, F., Crowell, S. M. R., Feng, L., Palmer, P. I., Dubey, M., García, O. E., Griffith, D. W. T., Hase, F., Iraci, L. T., Kivi, R., Morino, I., Notholt, J., Ohyama, H., Petri, C., Roehl, C. M., Sha, M. K., Strong, K., Sussmann, R., Te, Y., Uchino, O., and Velazco, V. A.: Improved retrievals of carbon dioxide from Orbiting Carbon Observatory-2 with the version 8 ACOS algorithm, Atmos. Meas. Tech., 11, 6539–6576, https://doi.org/10.5194/amt-11-6539-2018, 2018.

Taylor, T. E., O'Dell, C. W., Baker, D., Bruegge, C., Chang, A., Chapsky, L., Chatterjee, A., Cheng, C., Chevallier, F., Crisp, D., Dang, L., Drouin, B., Eldering, A., Feng, L., Fisher, B., Fu, D., Gunson, M., Haemmerle, V., Keller, G. R., … Zong, J.: Evaluating the consistency between OCO–2 and OCO-3 XCO2 estimates derived from the NASA ACOS version 10 retrieval algorithm. Atmospheric Measurement Techniques, 16(12). https://doi.org/10.5194/amt-16-3173-2023,2023.

This references are cited in the revised manuscript.

---

## Author Comment (AC2)

The paper is extremely poorly written and is unacceptable in its current form. While I believe the analyses are interesting, I highly recommend the authors to review the text and rewrite as needed. Apart from the grammatical errors, there are also several errors in the introduction- which I would expect the senior authors to have addressed. I provide some examples from the Abstract and Introduction. I would be happy to re-review this paper if the manuscript is resubmitted.

[Reply] We thank Dr. Rastogi for carefully reading our work and providing thoughtful comments. We have tried our best to significantly improve the writing and also done extra analysis. Our replies are in black below each comment in grey.

Because the revisions are very significant throughout the manuscript, a revised draft is supplied with this reply for your reference, which will be further improved when we are allowed to revise it.

Abstract:
line 18: We show the model...
line 20: comparisons reveal these differences
line 21: define the acronym ACTM when first used
line 24: unclear, because of course 4 km is not representative of XCO2 (which by definition is total column)
line 27: due to its course resolution and the higher resolution

[Reply] The abstract is rewritten completely. Hope it is easier to read now. Kindly check the revised draft.

Introduction:
line 44: majorly responsible for global warming

[Reply] We revised the sentence to: "*Atmospheric $CO_2$ is the most significant anthropogenic greenhouse gas (GHG) in the Earth's atmosphere, primarily responsible for global warming and climate change since the preindustrial era, circa 1750 (Bolin & Eriksson, 1959)*"

line 48: chemical transport model

[Reply] We changed the sentence to: "*These $CO_2$ measurements have very high accuracy (0.1 ppm in 400ppm) (WMO, 2020) and long-term records, making them widely used in Bayesian "top-down" estimation of surface $CO_2$ fluxes using atmospheric chemistry-transport models (Enting and Mansbridge, 1989; Tans et al., 1990; Gurney et al., 2002; Peylin et al., 2013)*"

line 52: perhaps data sparse is better than data void

[Reply] We have replaced "*void*" with "sparse" in the revised version, as suggested.

line 54: space-based satellites (not measurements). OCO-2 is a satellite that "measures" XCO2, OCO-2 itself is not a measurement

[Reply] We agree with your comment. We have replaced "measurements" with "*satellites*".

line 61: measurements have shown retrieval errors

[Reply] We revised the sentence to: "*OCO-2 version 10 retrievals of $XCO_2$ exhibit a mean bias (RMSE) of 0.24 (0.81) ppm over land and 0.43 (0.84) ppm over oceans globally when compared to the more accurate*

*WMO scale-maintained XCO$_2$ measurements (accuracy < 0.5 ppm) from surface-based Total Carbon Column Observation Network (TCCON) sites (Taylor et al., 2023)".*

line 64: Wunch et al., 2017 is cited for OCO b10 but this study came out before OCO-2 b10 which was released in 2020/2021.

[Reply] We agree with the reviewer's comment. We have removed the citation from the text and new information are provided for improving clarity about data used in the analysis.

line 66: CO$_2$ fluxes at regional scales
line 68: Millet et al., 2007 isn't a great reference for this because observing systems and models have both tremendously improved since that study. Current precision requirements are below 1 ppm for inferring regional fluxes. See Feng et al., 2019 (JGR-A) and Rastogi et al., 2021 ACP.
line 71: Massie et al., 2021 show the impact of 3D cloud radiative impacts on XCO2 (bias). This is different from cloud effects (which could mean no data over cloudy conditions).
line 73: estimates not estimations
line 75: This is incorrect. OCO-2 has always had a DEM. Jacobs et al., 2023 found that the choice of DEM lead to large systematic errors in high latitudes for OCO-2 b10.
line 77: transport not such kind
lines 78-80: needs to be re-written

[Reply] In the revised draft we have removed these unclear sentences from the introduction section to improve clarity and align better with the study's objectives, without compromising the article's content. Kindly refer to the revised draft as provided with this reply.

line 86: inversion estimates through...

[Reply] We changed the sentence to: "*Thus, assessing uncertainties linked with specific transport models through comparison of partial-column (layers of the atmosphere) CO$_2$ from near surface to upper troposphere using the aircraft CO$_2$ vertical profiles across the globe are desired.*"

lines 90-91: This needs to be clarified. Schuh et al., 2019 have examined the impact of transport related errors on flux estimates. Feng 2019 and Rastogi 20201 used aircraft profiles to understand combined flux and transport errors.

[Reply] Thank you for mentioning it. We find Rastogi et al. 2021 is a better reference because of direct use of OCO-2. Therefore, we changed the sentence to: "*Limited studies are available in literature to assess altitude-wise uncertainties of transport models using surface and aircraft CO$_2$ measurements over locations of OCO-2 samplings (Rastogi et al., 2021).*".

line 96: four specific aircraft sites

[Reply] We mentioned the site's name now. Revised text reads as "*four fixed sites (TEF, SAN, RBA, and ALF)*"

References:
Feng 2019: http://dx.doi.org/10.1029/2019JD031165
Rastogi 2021: https://doi.org/10.5194/acp-21-14385-2021
Schuh et al., 2019  10.1029/2018GB006086
Citation: https://doi.org/10.5194/egusphere-2024-3976-RC2

All these references are cited in the revised manuscript.